# OpenReview forum: "CoGe-GCD: Reframing Generalized Category Discovery with Compositional Generalization"
_ICML.cc/2026/Conference — ICML 2026 regular_

### Official Review · Reviewer_VZ29 · 2026-03-01

**Soundness:** 3
**Presentation:** 4
**Significance:** 3
**Originality:** 3
**Overall Recommendation:** 6
**Confidence:** 5

**Summary:**

This paper revisits Generalized Category Discovery (GCD) and argues that success in open-world discovery requires a form of compositional reasoning: the model reuses primitives learned from known classes and decides when novel combinations imply novel categories.  To operationalize this view, the authors propose CoGe-GCD, a plug-and-play inductive-bias module inserted between a ViT backbone and an existing GCD head, without modifying the head or the loss.   CoGe-GCD consists of two coupled stages.\
Compositional Perception maps patch tokens into a compact vocabulary of primitives via competitive token–primitive assignments and token–primitive information passing, producing refined token embeddings.  Generalizing Induction then applies a geometry-structure-preserving calibration that aligns token memberships with proximity and continuity while keeping column-stochastic probabilistic semantics.  Across standard fine- and coarse-grained benchmarks, the module delivers consistent improvements in overall and novel-category accuracy and improves unknown-class number estimation with marginal overhead.

**Compliance With Llm Reviewing Policy:**

Affirmed.

**Final Justification:**

My concerns have been addressed, I have no other questions.

**Key Questions For Authors:**

1.Are the low-rank and spatial smoothness assumptions intended mainly as analytical abstractions, or do they reflect properties of real ViT representations, and how should readers interpret their scope in practice?\
2.What key properties motivate the specific formulation of the geometric calibration operator, and why is this particular design well suited for generalized induction in GCD?\
3.Why do the empirical gains appear more pronounced in fine-grained settings than in coarse-grained ones, and what does this suggest about the scope of the method?

**Limitations:**

yes

**Strengths And Weaknesses:**

Strengths\
1.The paper targets a real gap in how the community frames GCD.   Instead of treating GCD as better clustering, the paper frames it as a process that first structures perceptual evidence into reusable primitives and then performs induction over that structure for known/novel decisions. This decomposition into Compositional Perception and Generalizing Induction gives a clear missing capability story and makes the paper easy to evaluate on its own terms.  \
2. Compositional Perception  design explicitly represents token-to-primitive memberships and uses token to primitive aggregation followed by primitive to token dissemination, with degree normalization to avoid domination by high-degree tokens/primitives.   This makes the intermediate representation interpretable as structured evidence, and it aligns with the paper's goal of organizing patch-level information before discovery objectives are applied. \
3. Generalizing Induction adds an inductive bias in a disciplined way, and the semantics-preserving aspect is a nice touch. This directly matches the stated goal of injecting proximity/continuity priors without breaking the probabilistic meaning of primitive competition, which is an appealing design principle for a plug-in module. \
4. The plug-and-play story is credible and practically valuable. The module keeps the backbone and GCD head unchanged and feeds the head a calibrated representation obtained by averaging refined tokens and adding to the class token.  This minimal interface is exactly what one wants if the contribution is meant to generalize across frameworks rather than win within a single bespoke pipeline. \
5. The main results show average improvements on fine-grained and coarse-grained benchmarks, including gains on novel categories, which fits the paper's narrative that structuring tokens helps discovery.  The hierarchical analysis reports better estimates of unseen category counts, and the overhead analysis suggests the added parameters/FLOPs and runtime are modest.\
Weaknesses\
1.The paper explicitly motivates GCD novelty as new combinations of primitives already present in known classes.  However, the experiments follow standard GCD benchmarks and protocols, which do not ensure that novel classes are primarily recombinations of shared primitives rather than genuinely new appearance modes. \
2. The model assumes token features are a low-rank linear composition of primitives plus isotropic Gaussian noise, with a spatial smoothness condition on assignments.  The main proposition relies on regimes where noise variance dominates the bias introduced by smoothing. Could the author elaborate on whether there might be scenarios where this hypothesis fails? I'd like to see more analysis.\
3. The module averages refined tokens and adds the result to the original class token before the GCD head.  This keeps the plug-and-play story simple, but it may discard useful structure that the refinement produces, especially for heads that could directly exploit token-level information.\
4. The paper evaluates unknown-class number estimation and shows improvements.  This is useful, but the setting still assumes clustering the full dataset with Hungarian matching under a fixed protocol.  It would be valuable to see at least one more realistic stressor to support the plug-and-play in open-world pipelines framing.

---

> ### Author Rebuttal · Authors · 2026-03-31
>
> We sincerely thank the reviewer for the insightful comments and the positive assessment of our framing of GCD through **Compositional Perception** and **Generalizing Induction**. We respond briefly below.
>
> > W1 (about primitive recombination)
>
> We agree that standard benchmarks are not explicitly constructed to enforce "known-primitives recombination." To address this concern, we provide stronger supporting evidence:
>
> We provide additional primitive visualizations at [https://anonymous.4open.science/r/CoGe-GCD](https://anonymous.4open.science/r/CoGe-GCD). On fine-grained datasets, primitives with the same index often attend to semantically consistent regions (e.g., windows, headlights, wings). On coarse-grained datasets, they more often capture higher-order correlations.
>
>
> > W2 & Q1 (about the low-rank and spatial smoothness assumptions)
>
> Our view is that these assumptions are **primarily analytical abstractions**, introduced to make the two-stage design theoretically interpretable, rather than literal claims about all ViT features.
>
> * The proposition is already **conditional**: the benefit holds when denoising outweighs the bias introduced by smoothing.
> * In practice, the assumptions describe the regime where CoGe-GCD is most helpful: token features contain reusable low-dimensional structure, and nearby tokens tend to share related primitive assignments.
> * They may weaken under highly non-local dependencies, severe clutter/occlusion, or when smoothing bias dominates denoising gain.
>
> We will clarify this scope and add further analysis in the revised appendix.
>
>
>
> > W3 (about token-level structure)
>
> Our choice is deliberate: CoGe-GCD is designed as a **drop-in inductive-bias module**, so we keep both the backbone and the GCD head unchanged.
>
> * The simple token-averaging interface prioritizes **universality and compatibility** across very different GCD frameworks.
> * A specially designed token-level head could potentially exploit more of the refined structure; we view this as a promising extension rather than a contradiction to the current design goal.
>
> > W4 (more realistic stress test)
>
> In the main text, we evaluate **unknown-class number estimation** and clustering **without ground-truth (K)**. To further address this concern, we additionally construct a realistic biomedical scenario: **generalized rare-disease category discovery from common-disease supervision**, where rare diseases are treated as novel classes.
>
> We evaluate on **Endoscopy, Microscopy, and Dermatology**. As shown in **Table R2**, CoGe-GCD substantially outperforms the strong baseline, improving both known-class performance and discovery of **novel rare diseases**. We will add these results in the revised paper.
>
> **Table R2: Comparison on real-world setting**
>
> | Methods | Endo (A/K/N) | Micro (A/K/N) | Derm (A/K/N) |
> | :--- | :--- | :--- | :--- |
> |  SelEx | 50.1/52.0/47.4 | 66.4/68.3/61.3 | 28.9/29.1/27.9 |
> | **SelEx+Ours** | **71.2/83.4/53.0** | **80.7**/**89.3**/58.6 | **35.6/35.6/35.6** |
>
>
> *\*Endo=Endoscopy, Micro=Microscopy, Derm=Dermatology.*
>
> > Q2 (geometric calibration operator)
>
> The operator is designed to satisfy three goals at once:
> * Inject **proximity** and **continuity**, since primitive competition alone only uses feature affinity;
> * Remain **lightweight** and easy to integrate;
> * Preserve the **probabilistic semantics** of primitive competition.
>
> This is why Eq. (11) defines a spatial kernel, Eq. (12) recalibrates memberships with renormalization, and Eq. (13) blends calibrated and original memberships instead of overwriting them.
>
> > Q3 (why are gains larger in fine-grained settings)
>
> We believe this pattern matches the intended mechanism of CoGe-GCD.
> * Fine-grained categories usually share more local primitives and differ through subtle recombinations of parts/attributes.
> * Therefore, explicit primitive organization and calibrated geometry have more room to help.
> * Coarse-grained categories often differ through larger global appearance gaps, so the benefit is still present but smaller.
>
> This suggests that CoGe-GCD is effective when discovery depends on shared primitive structure and subtle recombination.
>
> ---
>
> We sincerely thank the reviewer for the high recognition. We hope that the added visualizations, realistic biomedical stress test, and clearer explanation of the theoretical scope and design trade-offs will help the reviewer better understand the mechanism of CoGe-GCD. Should you have any other questions, we warmly welcome any further in-depth discussions.

---

> > ### Author Rebuttal · Reviewer_VZ29 · 2026-04-03
> >
> > My concerns have been addressed, I have no other questions.

---

> > > ### Author Response · Authors · 2026-04-04
> > >
> > > Dear Reviewer VZ29,
> > >
> > > We are grateful for your positive feedback and glad to hear that our response **adequately addressed your concerns**. Thank you for your valuable time and support.

---

### Official Review · Reviewer_g2tn · 2026-03-07

**Soundness:** 2
**Presentation:** 3
**Significance:** 3
**Originality:** 3
**Overall Recommendation:** 4
**Confidence:** 3

**Summary:**

This paper proposes CoGe-GCD, a novel inductive-bias module for the Generalized Category Discovery (GCD) task. Grounded in the Latent Compositional Manifold Assumption, the work decouples the discovery process into two coupled stages: first, Compositional Perception, which maps unstructured tokens to a reusable vocabulary of primitives through primitive competition and evidence consolidation; and second, Generalizing Induction, which leverages the induced geometric structure and applies geometric calibration based on the principles of proximity and continuity. The module is designed to be plug-and-play, allowing integration into various existing GCD frameworks.

**Compliance With Llm Reviewing Policy:**

Affirmed.

**Final Justification:**

The authors have provided additional experiments and visualizations that address most of my concerns; therefore, I will keep positive toward the paper.

**Key Questions For Authors:**

1. I am curious about that what is the specific role of the additional parameters introduced by CoGe-GCD? What is the ratio of these introduced parameters to the total trainable parameters? If the parameter count is significant, could the performance gains partially stem from the increased model scale?
2. What is the actual value used for the temperature coefficient ($\tau$) in Equation 5? Does this value require dataset-specific tuning (e.g., between CIFAR-100 and ImageNet-100) to maintain the effectiveness of primitive competition?
3. Is there any visual evidence or qualitative analysis to support the motivation of "recombining primitives", as you illustrated in Fig.2?
4. How is the spatial proximity kernel ($\Pi$) in Equation 11 defined? Does this operator possess enough adaptivity when applied to non-square images or varying patch-sequence resolutions?

**Limitations:**

No, it currently lacks a dedicated section on limitations and societal impact. The proposed method does not have any obvious negative societal impacts. A potential limitation of the approach is that as image resolution (and thus the number of image tokens) increases, the additional computational overhead introduced by the proposed module may become increasingly unacceptable.

**Strengths And Weaknesses:**

Strengths:

1. The paper introduces human-like reasoning (reusing known primitives and inducing new combinations) into the GCD task, providing an insightful perspective for open-world category discovery.

2. The proposed CoGe-GCD module is highly versatile. Experimental results demonstrate that adding this module to several previously representative methods (such as SimGCD, LegoGCD, CMS, and SelEx) brings performance improvements across multiple benchmarks, proving its effectiveness.

3. The Latent Compositional Manifold Assumption provides a mathematical basis for the methodology, and Proposition 3.1 (along with its proof) offers theoretical guarantees for reducing reconstruction error and suppressing noise through this two-stage process.

Weaknesses:

1. The paper introduces a temperature coefficient ($\tau$) in Equation 5 to control the degree of primitive competition, but neither its specific value nor an ablation study is provided.

2. Slight degradation in Known Class performance: Observation of the experimental results (Table 1) reveals that while All Accuracy improves, the performance on known classes (Known Acc) slightly decreases in certain settings (e.g., the SimGCD and CMS baselines on the CUB-200 dataset). This suggests that the module might interfere with the discriminative power of the original features for known classes while enhancing generalization for novel classes, a trade-off that is not thoroughly discussed.

---

> ### Author Rebuttal · Authors · 2026-03-31
>
> We sincerely thank the reviewer for the positive assessment of our motivation, versatility, and theoretical framing, as well as for the constructive questions. Below we address each concern and will incorporate the clarifications in the revised paper.
>
>
> > W1 & Q2 (temperature coefficient in Eq. 5)
>
> In our implementation, the temperature in Eq. (5) is fixed to $\tau=1$.
>
> * We include $\tau$ solely for formal consistency with the standard softmax formulation.
> * We do **not** tune $\tau$ across datasets, as CoGe-GCD is designed as a **plug-and-play** module with minimal hyperparameter sensitivity.
> * Thus, $\tau$ is not intended as an additional dataset-specific degree of freedom on which the method depends.
>
>
>
> > W2 (slight Known drop in a few settings)
>
> * GCD inherently involves an open-world **known/novel trade-off**: improving the separability of novel categories can slightly reshape the latent space that was previously favorable to known classes.
> * However, on the cited CUB-200 cases, this trade-off is minimal: the Known drops are only **-0.9** and **-0.3**, while the Novel gains are substantial (**+3.8** and **+2.8**).
> * To preserve the **universality** of CoGe-GCD, we intentionally avoid dataset-specific tuning. As shown in **Table R1**, if the number of primitives $M$ is adjusted specifically for each dataset, both Known and Novel accuracies can improve simultaneously. We chose to report one unified setting for fairness. We will add this discussion in the revision.
>
> **Table R1: Ablation on varying numbers of primitives ($M$) with SimGCD on CUB dataset**
> | Setting | All | Known | Novel |
> | :--- | :--- | :--- | :--- |
> | SimGCD | 61.6  | 66.4 | 59.3 |
> | + Ours ($M=8$) |  62.7  | 64.1 | 62.1 |
> | + Ours ($M=16$) |  63.9  | 65.5 | 63.1 |
> | + Ours ($M=32$) |  **64.5**  | **67.8** | **62.8** |
>
>
> > Q1 (what do the extra parameters do, and are the gains from model scale?)
>
> * They originate mainly from the primitive bank $P_0$ in Eq. (4) and the lightweight message-passing matrices.
> * Their sole purpose is to perform **primitive adaptation** and **evidence consolidation**, not to scale the backbone or GCD head.
> * As shown in Table 4, the model size increases from 92.10 MB to 92.16 MB (**<0.1% additional parameters**), with marginal FLOPs and runtime overhead.
> Therefore, the substantial gains stem from our compositional structuring and calibrated message passing, not naive model scaling.
>
> > Q3 (evidence for “recombining primitives”)
>
> Beyond Fig. 6, we now provide additional primitive visualizations at [Link/CoGe-GCD](https://anonymous.4open.science/r/CoGe-GCD).
> * On **fine-grained datasets**, primitives with the same index consistently attend to semantic regions (e.g., windows, wings).
> * On **coarse-grained datasets**, they capture realistic **higher-order correlations**.
>
> To quantitatively verify this, we perform **primitive masking** at test time (**Table R2**). Masking semantically meaningful primitives causes clear degradation, especially on **novel classes**, directly proving that CoGe-GCD relies on reusable primitive structure to discover novel categories.
>
> **Table R2: Comparison on Primitive Masking with SelEx**
> | Methods | CUB (A/K/N) | Air (A/K/N) | S-Cars (A/K/N) |
> | :--- | :--- | :--- | :--- |
> | SelEx | 75.6/77.3/74.7 | 61.1/68.7/57.3 | 55.5/77.6/44.8 |
> | SelEx + Ours  | **80.9/81.2/80.7** | **62.9/67.4/60.6** | **60.3/80.8/50.4** |
> | w/o Prim #1 | 78.9/80.7/78.1 | 61.1/66.2/58.6 | 56.8/78.9/46.1 |
> | w/o Prim #6 | 78.5/81.0/77.3 | 61.7/67.2/59.0 | 57.6/77.9/47.8 |
>
> > Q4 (definition and adaptivity of the spatial proximity kernel in Eq. 11)
>
> The kernel $\Pi$ is a **row-stochastic Gaussian kernel** over token positions.
> * It uses normalized spatial coordinates $p_i \in \mathbb{R}^2$ to perform one-step local spatial diffusion, weighting nearby tokens higher to enforce **proximity** and **continuity**.
> * Because it relies on normalized coordinates rather than absolute pixels or a fixed grid, it naturally adapts to **non-square images** and **variable patch resolutions**.
> * The re-normalization in Eq. (12)–(13) ensures the calibrated memberships retain their column-stochastic primitive-competition semantics. We will explicitly clarify this adaptivity in the revised paper.
>
> > Limitations / Societal Impact
>
> We will add a dedicated section in the revised paper. We will explicitly discuss that while the overhead is currently marginal, scaling to significantly larger token counts or higher resolutions may increase computational costs. We will also clarify the method's scope, confirming it introduces no obvious new high-risk capabilities beyond standard GCD pipelines.
>
> ---
>
> We sincerely appreciate the reviewer for these constructive comments. We hope these clarifications, together with the added visualizations, primitive-masking analysis, and explicit implementation details, address the reviewer‘s concerns. If anything remains unclear, we warmly welcome further in-depth discussions.

---

> > ### Author Rebuttal · Reviewer_g2tn · 2026-04-01
> >
> > The authors have responded and clarified raised questions.

---

> > > ### Author Response · Authors · 2026-04-04
> > >
> > > Dear Reviewer g2tn,
> > >
> > > We sincerely thank you for your recognition of our work and for confirming that our response has **adequately addressed your concerns**. We deeply appreciate your time and support.

---

### Official Review · Reviewer_A1sc · 2026-03-11

**Soundness:** 2
**Presentation:** 2
**Significance:** 2
**Originality:** 2
**Overall Recommendation:** 3
**Confidence:** 4

**Summary:**

The paper studies Generalized Category Discovery (GCD), which aims to group test images: (1) if images are from a specific pre-defined category, they should be grouped; (2) otherwise, they should be clustered into discrete groups hypothetically belonging to some unseen categories. To address GCD, the paper argues that a GCD method should conduct compositional reasoning like human, reusing primitives learned from known classes and deciding when new combinations imply new categories. To model this compositional reasoning process, the paper proposes CoGe-GCD, consisting two stages: Compositional Perception, and Generalizing Induction. The former structures patch tokens by mapping them to a small vocabulary of primitives and refining token embeddings via competitive token-primitive assignment and information passing. The latter exploits the induced geometric structure and applies a geometric-structure-preserving calibration over spatial relations, maintaining probabilistic semantics while improving extrapolation to unseen primitive combinations. It explains that CoGe-GCD is an inductive-bias module between backbone and projection head, without modifying heads or losses, and can be plugged into diverse GCD frameworks. On standard benchmarks, it shows CoGe-GCD achieves improved performance w.r.t various metrics.

**Compliance With Llm Reviewing Policy:**

Affirmed.

**Final Justification:**

After carefully considering the authors' rebuttal, I would at best raise my rating to Weak Reject. The following concerns prevent me from positively rating this paper.


**Significance of GCD**.
I have carefully considered the real-world relevance of GCD and encouraged the authors to articulate concrete application scenarios where GCD is actively employed. While the rebuttal mentions domains like medical diagnosis and autonomous driving, the concrete connection between these domains and GCD remains unclear and appears speculative. Although the broader relevance of GCD as a research problem is not solely determined by this paper, I believe it is important for the paper to clearly and pedagogically demonstrate how GCD aligns with practical, real-world applications rather than remaining an abstract benchmark.

**Experimental protocol of GCD**.
I knew well the problem of GCD (as well as open set, open world and so on) and understood that the literature started with simulated experiments and (artificial) datasets. GCD papers including this one repurpose existing datasets (designed for image classification)  by withholding certain object classes as novel ones. This setup is artificial while convenient, limiting the practical relevance of proposed GCD methods. The rebuttal provided additional results on new datasets but presented little details about the data, making it difficult to assess the validity and significance of the results.

**Visualizations and Supplemental Code**.
The figures in the paper are difficult to interpret due to readability issues. The supplement contains a single python script which did not have comments and did not implement all the called functions. Although the rebuttal provided an anonymous website that listed new figures and code, the incomplete and delayed nature of these materials undermines confidence in the transparency and rigor of the submission. For this, I am unable to convince myself to raise my rating further.

**Key Questions For Authors:**

Key questions are centered at the weaknesses listed above.

1. While GCD seems like an interesting problem that combines open-set recognition and clustering, it is unclear what real-world applications can benefit from this research? The datasets used in experiments are quite artificial that do not reflect real-world applications. Can authors discuss on this?

2. The most recent benchmarked methods (CMS, LegoGCD, SelEx, InfoSieve, AMEND and uGCD) were published in 2024 (two years ago). As this paper is submitted to 2026's ICML, it is natural to ask whether there are more recent published works in GCD. If yes, how they perform on the standard datasets. If no, why does the community not study GCD?

3. Can authors demonstrate what primitives the model has learned and how the model finds data of novel classes based on novel combinations of the learned primitives? The demonstration is preferably through visualization.

4. Regarding to Weaknesses 4 and 5, authors can try to address them in rebuttal.

**Limitations:**

The paper does not discuss limitations and societal impact. Authors can think about potential societal impact and limitations pertaining to the research protocol of GCD and the developed method.

**Strengths And Weaknesses:**

**Strengths**
1. GCD looks like an interesting problem that combines open-set recognition (classifying an image into one of K predefined classes or an unknown class), and clustering (grouping images to potentially align oracle class IDs).

2. The intuition of modeling human-like compositional reasoning process makes sense.




**Weaknesses**
1. The datasets in experiments datasets are artificial, e.g., splitting a dataset (bird species in CUB-200, aircraft models in FGVC Aircraft, etc.) into known classes and unknown classes. That said, the datasets do not reflect real-world scenarios, although they might be widely used in the (previous?) GCD literature. The paper is expected to use appropriate datasets to realistically simulate the current GCD protocol, e.g., in what applications the number of known and unknown classes is given, unlabeled data need to be classified and grouped (according to potential categories).

2. Compared methods were published two years ago and it is unclear how GCD has been addressed recently. Without this context, it is hard to evaluate the significance.

3. There are insufficient visualizations to demonstrate the effectiveness of compositional reasoning by the proposed CoGe-GCD method. Without such, it is unclear (1) how the learned primitives (parts, attributes, relations) look like, and (2) whether CoGe-GCD relies on novel combinations of known primitives to discover novel categories of test images.

4. The submission includes a supplementary zip file which contains a single python script but does not implement all functions called therein. This incomplete, imcomprehensive python file does not serve as supplementary material to support the paper but hurt the accountability of the paper. Hence, the reviewer thinks this as a major weakness.

5. The readability of the paper needs to be improved. For example, Figure 1, 4, 5 are too small to read. Figure 2 is not self-explanatory, e.g., it is unclear what dots on the dog image mean - are they detected primitives? The paper does not explain evaluation protocols such as metrics and validation.

---

> ### Author Rebuttal · Authors · 2026-03-31
>
> We appreciate the reviewer's recognition of the human-like compositional reasoning process at the core of CoGe-GCD and the thoughtful feedback. We address each concern below.
>
> > W1 & Q1 (about benchmarks and real-world relevance)
>
> We agree standard GCD benchmarks are controlled proxies rather than full real-world deployments. Our original paper follows the canonical GCD protocol mainly for fair comparison. To directly address the reviewer’s concern, we additionally build two realistic biomedical settings:
> * **Setting I:** Generalized rare-disease discovery from common-disease supervision (rare diseases = novel classes).
> * **Setting II:** Abnormal diseased-pattern discovery from normal organ/tissue supervision (pathological categories = novel classes).
>
> We evaluate on **Endoscopy, Microscopy, and Dermatology**. As summarized in **Table R1**, CoGe-GCD substantially outperforms the strong baseline **SelEx** across both settings. Importantly, CoGe-GCD improves not only known categories but also the discovery of **novel rare/pathological categories** with minor fluctuation, showing that primitive-based compositional perception remains highly effective in real-world scenarios.
>
> **Table R1: Real-World Biomedical Discovery (Settings I & II)**
> | Setting | Methods | Endo (A/K/N) | Micro (A/K/N) | Derm (A/K/N) |
> | :--- | :--- | :--- | :--- | :--- |
> | **I: Rare Disease** | SelEx | 50.1/52.0/47.4 | 66.4/68.3/61.3 | 28.9/29.1/27.9 |
> | | **SelEx+Ours** | **71.2/83.4/53.0** | **80.7/89.3**/58.6 | **35.6/35.6/35.6** |
> | **II: Abnormal** | SelEx | 50.3/51.9/41.7 | 58.5/58.0/59.8 | 29.3/28.2/40.7 |
> | | **SelEx+Ours** | **70.1/86.0/46.3** | **87.1/89.1/81.7** | **33.6/33.3**/36.4 |
>
> *\*A/K/N denotes All/Known/Novel Accuracy (%).*
>
> > W2 & Q2 (about baselines)
>
> Our baselines are chosen to cover **representative GCD paradigms**:
> * **SimGCD:** prototype-based. **LegoGCD:** regularization-based. **CMS:** contrastive. **SelEx:** hierarchical pseudo-labeling.
> These are strong, recognized baselines, chosen to validate plug-and-play **generality** rather than optimizing for one specific framework.
>
> To further enhance comparisons, we included newer methods from CVPR 2025 (HypGCD). As shown in **Table R2**, CoGe-GCD improves novel class discovery while maintaining competitive known class accuracy. This supports our claim that the compositional inductive bias is broadly effective across different optimization recipes.
>
> **Table R2: Comparison with new GCD Methods**
> | Methods | CUB (A/K/N) | Air (A/K/N) | S-Cars (A/K/N) |
> | :--- | :--- | :--- | :--- |
> | HypGCD | 77.5/75.2/78.7 | 64.3/68.4/62.3 | 58.8/78.0/49.6 |
> | HypGCD+Ours | **78.6**/73.5/**81.2** | **66.9/69.2/65.7** | **60.4/78.8/51.7** |
>
>
> > W3 & Q3 (visualization and primitive recombination)
>
> We provide additional visualizations at [Link/CoGe-GCD](https://anonymous.4open.science/r/CoGe-GCD). We observe:
> * On **fine-grained datasets**, the same primitive index often attends to semantically consistent regions (e.g., windows, headlights).
> * On **coarse-grained datasets**, primitives capture higher-order correlations rather than named parts, which is realistic for complex images.
>
> To directly test reliance on primitive recombination, we perform **primitive masking** at test time (**Table R3**). Masking semantically meaningful primitives causes clear degradation, especially on **novel classes**, which perfectly aligns with our intended mechanism.
>
> **Table R3: Comparison on Primitive Masking with SelEx**
> | Methods | CUB (A/K/N) | Air (A/K/N) | S-Cars (A/K/N) |
> | :--- | :--- | :--- | :--- |
> | SelEx | 75.6/77.3/74.7 | 61.1/68.7/57.3 | 55.5/77.6/44.8 |
> | SelEx + Ours  | 80.9/81.2/80.7 | 62.9/67.4/60.6 | 60.3/80.8/50.4 |
> | w/o Prim #1 | 78.9/80.7/78.1 | 61.1/66.2/58.6 | 56.8/78.9/46.1 |
> | w/o Prim #6 | 78.5/81.0/77.3 | 61.7/67.2/59.0 | 57.6/77.9/47.8 |
>
> > W4 & Q4 (supplementary code)
>
> Our intention is to provide the core code for quick inspection. We now provide the **complete codebase and README** at [Link/CoGe-GCD](https://anonymous.4open.science/r/CoGe-GCD). We will ensure the revised supplementary package is complete and properly documented.
>
> > W5 & Q4 (readability, figure clarity, protocol explanation)
>
> * We agree that several figures are too small.
> * The dots on the dog image in Fig. 2 visualize primitives; we will make this explicit.
> * We will **redraw/re-layout Figs. 1/2/4/5**, enlarge fonts, and improve annotations for readability.
> * Regarding protocol details, our evaluation follows the **standard GCD protocol**  (in the supplementary materials of the original GCD paper). We agree these details should be more explicit and will provide a clearer explanation in the revised supplement.
>
> ---
>
> We sincerely appreciate the reviewer for these valuable comments. We hope the new realistic settings, baselines, masking analysis, and complete code release address reviewer's concerns. We warmly welcome any further discussions.

---

> > ### Author Rebuttal · Reviewer_A1sc · 2026-04-02
> >
> > Thanks for the rebuttal. The rebuttal provides new results on new datasets with a new compared method (HypGCD CVPR'25), aiming to address my concerns on artificial datasets, dated compared methods. Yet, the datasets based on Endoscopy, Microscopy, and Dermatology also simulate and do not mirror real-world applications -- do these application really use or need GCD?
> >
> > Moreover, supplementary materials and readability of figures were expected to be ready by the paper submission deadline, instead of being made up in rebuttal.
> >
> > In sum, I believe that fully addressing my concerns needs substantial work but I'll consider to adjust my rating.

---

> > > ### Author Response · Authors · 2026-04-04
> > >
> > > We sincerely thank you for the prompt response. We address your remaining concerns regarding **(1) the practical relevance of GCD** and **(2) figure sizes and the scope of the released code in the supplementary materials** below, noting that neither requires substantial technical revisions to the paper."
> > >
> > > > **Q1: Do these applications really use or need GCD?**
> > >
> > > **Yes.** While standard benchmarks are curated proxies, GCD's *problem formulation* is neither artificial nor niche. It addresses a critical bottleneck in open-world machine learning.
> > >
> > > **1. Fundamental positioning: from human annotation to autonomous discovery**
> > >
> > > Traditional supervised learning relies heavily on exhaustive human annotation. However, real applications operate in open environments with undefined categories and unavailable labelled data. Standard methods fail here, lacking the ability to structure data automatically and understand new concepts without external supervision.
> > >
> > > GCD drives a necessary paradigm shift. Instead of a fully unsupervised approach that reduces to a pure clustering problem, GCD utilizes known labelled instances to learn the foundational attributes constituting a semantic class.
> > >
> > > This structural knowledge is then transferred to unlabelled data to mathematically organize unknowns and discover new categories. By bridging known semantics with unknown structures, GCD transforms AI from a passive pattern matcher into an active engine for **Autonomous Knowledge Discovery**.
> > >
> > >
> > > > **2. Core ML challenge: bridging the representational gap**
> > >
> > > This positioning exposes a profound ML theoretical challenge: GCD highlights the dilemma between *continuous feature representations* and *discrete concept emergence*.
> > >
> > > Representation learning optimizes continuous latent spaces via known distributions, but discovering novel categories requires new discrete semantic boundaries. Generic geometric clustering fails as real-world novelties are complex variations of known distributions, not isolated points.
> > >
> > > Resolving GCD answers a fundamental ML question: *how can we mathematically induce new discrete semantic concepts from unstructured continuous representations?* Bridging this gap makes GCD a foundational, highly challenging ML problem.
> > >
> > >
> > > > **3. Real-world driving force: propelling ML towards the open world**
> > >
> > >
> > > This fundamental "autonomous knowledge discovery" capability propels ML from static closed-set environments to dynamic open-world deployments, with GCD as its core driver in reality:
> > >
> > >
> > > * **Case A: Precision Medicine & Novel Subtype Discovery**
> > >   * **Clinical Necessity:** We define "Known" as common annotated diseases and "Novel" as sparse uncatalogued variations. Hospitals possess ample "Known" data, but "Novel" anomalies are frequently misclassified as noise. GCD is indispensable to transcend passive rejection and organize these heterogeneous cases into coherent semantic clusters. Details: [Link](https://anonymous.4open.science/r/CoGe-GCD-Rebuttal).
> > >   * **The GCD Driving Force:** Beyond current ML limits, GCD enables Autonomous Knowledge Discovery and Open-World Taxonomy Generation. Inducing new concept emergence, it facilitates novel tumor subtype identification, turning ML into an active driver for targeted therapies with minimal human relabeling.
> > >
> > >
> > > * **Case B: Autonomous Driving and Long Tail Perception**
> > >   * **Real world Necessity:** Defining "Known" as standard traffic and "Novel" as emerging elements, GCD is indispensable. It transcends passive emergency braking by actively clustering recurring unknown trajectories into coherent concepts.
> > >   * **The GCD Driving Force:** Surpassing ML limits, GCD drives **Autonomous Knowledge Discovery**. Inducing concept emergence enables systems to catalog the open world, turning passive anomaly detection into an active data flywheel for continuous perception.
> > >
> > > In summary, GCD is the foundational formulation for open world knowledge expansion. Our work addresses this fundamental ML problem, enabling robust autonomous discovery across critical domains.
> > >
> > >
> > > > **Q2: About only core code in supplementary materials and readability of figures**
> > >
> > > We fully agree on maintaining a high bar for clarity and thank you for your helpful suggestions.
> > >
> > > **Figures:** While axis and legend texts in Figures 1, 4, & 5 were smaller than ideal, the key visual trends clearly convey our main findings. Increasing font size is a straightforward presentation fix, not a substantial revision to the paper. Updated figures: [Link](https://anonymous.4open.science/r/CoGe-GCD-Rebuttal).
> > >
> > > **Code:** We initially provided core scripts to help reviewers quickly inspect our key mechanism without navigating a massive codebase. Our intention was clarity, not omission. Recognizing the value of inspecting the complete pipeline, we have provided the full codebase.
> > >
> > >
> > > ---
> > >
> > > We sincerely thank you for your constructive feedback and hope these clarifications fully address your concerns and earn your support.

---

### Official Review · Reviewer_oxwB · 2026-03-13

**Soundness:** 3
**Presentation:** 2
**Significance:** 3
**Originality:** 3
**Overall Recommendation:** 5
**Confidence:** 2

**Summary:**

The paper tackles Generalized Category Discovery (GCD), where models assign unlabeled instances to known or novel categories, requiring compositional reasoning. Existing methods struggle with unseen compositions due to unstructured token features. The paper propose CoGe-GCD, which introduces a plug-in module with two stages: Compositional Perception, mapping tokens to a small primitive vocabulary to form coherent groups, and Generalizing Induction, calibrating geometric structure to better handle novel combinations. CoGe-GCD improves classification accuracy, unknown-category estimation, and geometric quality across benchmarks, with minimal computational overhead.

**Compliance With Llm Reviewing Policy:**

Affirmed.

**Final Justification:**

Method was technically sounds and rebuttal addressed my concerns. So I will maintain my positive score as well.

**Key Questions For Authors:**

1) Further clarification on points 1–3 in the weaknesses section.
2) The limitations and failure cases of the proposed approach have not been adequately addressed. I am particularly interested in the model's ability to handle complex scenes and if the proposed method can be extend to estimating new categories.

**Limitations:**

The manuscript would benefit from a discussion of potential limitations, failure cases, and possible negative societal impacts of the proposed approach.

**Strengths And Weaknesses:**

Strengths

1) I find the idea of introducing CoGe-GCD as a plug and play module to be a novel contribution.

2) The experimental study is conducted on standard benchmarks and shows promising performance compared to all considered baselines. The ablation study explores the contribution from each introduced module.

3) The paper is well written.

Weakness

1) A quantitative ablation examining the individual impact of $P_0$ and $\Delta P$ would help clarify their respective roles in the proposed primitive adaptation mechanism.

2) For clarity, Figure 3 could include more detailed labels or annotations indicating how each component maps to the modules described in the method section.

3) The method section is somewhat difficult to follow, and additional restructuring could improve readability.

---

> ### Author Rebuttal · Authors · 2026-03-31
>
> We sincerely thank the reviewer for the positive assessment of the paper’s novelty, empirical promise, and writing quality. We also appreciate the helpful suggestions on ablations, figure clarity, and limitations. We address each point below.
>
> > W1 & Q1 (clearer quantitative ablation for the primitive adaptation mechanism)
>
> Thank you for this valuable suggestion. We agree that a finer-grained ablation makes the primitive adaptation mechanism clearer. We isolate the two terms in Eq. (4): the learnable global primitive bank $P_0$ (dataset-level prior) and the image-conditioned offset $\Delta P$ (image-specific adjustment). We also compare whether $\Delta P$ is derived from $avg$ or $max$ pooling (**Table R1**). The results indicate:
> * The best performance is achieved when both $P_0$ and $\Delta P$ are used.
> * A flexible learnable $P_0$ is essential for stable base accuracy.
> * $\Delta P$ helps preserve category-specific accuracy, particularly for known classes.
> * $avg$ pooling is generally more beneficial, likely because it captures stable discriminative token statistics, while $max$ pooling may introduce sparse but unstable cues.
>
> **Table R1: Ablation on $P_0$ and $\Delta P$ with SelEx**
> | Components | CUB (A/K/N) | Air (A/K/N) | S-Cars (A/K/N) |
> | :--- | :--- | :--- | :--- |
> | SelEx | 75.6/77.3/74.7 | 61.1/68.7/57.3 | 55.5/77.6/44.8 |
> | SelEx + CoGe-GCD | **80.9/81.2/80.7** | **62.9/67.4/60.6** | **60.3/80.8/50.4** |
> | w/o $P_0$ | 79.1/80.0/78.6 | 61.9/67.9/58.9 | 59.3/79.0/49.8 |
> | w/o $\Delta P$ | 79.5/78.3/80.1 | 61.1/67.0/58.2 | 58.7/78.1/49.3 |
> | w/o $max \ pool$ | 80.2/80.3/80.2 | 61.3/67.3/58.4 | 59.5/79.7/49.7 |
> | w/o $avg \ pool$ | 79.7/79.5/79.8 | 62.2/67.1/59.8 | 58.6/80.1/48.2 |
>
> *\*A/K/N denotes All/Known/Novel Accuracy (%). Results are reported on fine-grained benchmarks.*
>
> We will add these results and discussion in the revised paper.
>
>
>
> > W2 & Q1 (Fig. 3 should more explicitly correspond to the method)
>
> Thank you for this helpful comment. Fig. 3 is intended as an **intuitive illustration** of the principles of **proximity** and **continuity**. In the actual method, these principles are instantiated by Eq. (11)–(13):
> * Eq. (11) defines a row-stochastic spatial proximity kernel over normalized token coordinates.
> * Eq. (12)–(13) use it to calibrate token-to-primitive memberships while preserving column-stochastic competition semantics.
>
> Essentially, Fig. 3 provides the perceptual intuition, while Eq. (11)–(13) provide the operational realization. We agree this bridge should be more explicit. In the revised version, we will add clearer labels/arrows and directly annotate the correspondence from Fig. 3 to Eq. (11)–(13).
>
>
> > W3 & Q1 (the method section is somewhat difficult to follow)
>
> We appreciate this suggestion. The current presentation is compact due to page limits. In the revised version, we will reorganize Section 3 more clearly. We will add a short roadmap paragraph at the beginning and make the correspondence between **primitive adaptation**, **competition**, and **geometric calibration** more explicit. We will also provide a simplified flow diagram to make the pipeline easier to follow.
>
> > Q2 (limitations, failure cases, complex scenes, and new-category estimation)
>
> Thank you for raising this point. We agree these aspects should be discussed more explicitly.
> * **New-category estimation:** This is part of our evaluation. Table 2 shows that CoGe-GCD improves unknown-class number estimation.
> * **Limitations:** Two key limitations exist: (1) gains are more pronounced on fine-grained datasets where categories share reusable local primitives; (2) in some settings, Known accuracy decreases slightly due to the open-world trade-off and our choice to remain plug-and-play.
> * **Complex scenes:** To address this, we build a realistic biomedical setting: **discovering rare diseases (Novel) from common diseases (Known)**.
>
> **Table R2: Comparison on real-world biomedical discovery**
> | Methods | Endo (A/K/N) | Micro (A/K/N) | Derm (A/K/N) |
> | :--- | :--- | :--- | :--- |
> |  SelEx | 50.1/52.0/47.4 | 66.4/68.3/61.3 | 28.9/29.1/27.9 |
> | **SelEx+Ours** | **71.2/83.4/53.0** | **80.7**/**89.3**/58.6 | **35.6/35.6/35.6** |
>
> *\*A/K/N denotes All/Known/Novel Accuracy (%). This setup evaluates the model's ability to handle high-stakes, noisy clinical data.*
>
> We will add these discussions and results in the revised paper, alongside a dedicated **limitations/failure cases** section. We will also provide detailed experimental settings (dataset splits, hyperparameters) to demonstrate the real-world applicability of this task.
>
> ---
>
> We sincerely appreciate the reviewer's valuable comments and recognition. We hope the added results and clarifications address the concerns. If any doubts remain, we eagerly welcome further discussions.

---

> > ### Author Rebuttal · Reviewer_oxwB · 2026-04-04
> >
> > The response satisfactorily addresses most of my concerns. Taking into account the points raised  by other reviewers, my overall evaluation remains unchanged and I will maintain the same score.

---

> > > ### Author Response · Authors · 2026-04-04
> > >
> > > Dear Reviewer oxwB,
> > >
> > > Thank you for reviewing our rebuttal and confirming that your concerns **have been satisfactorily addressed**. We deeply appreciate your time and your positive evaluation.

---

### Decision · Program_Chairs · 2026-04-30

**Decision:**

Accept (regular)

**Comment:**

This paper proposes CoGe-GCD, an inductive-bias module for the Generalized Category Discovery (GCD) task. Inspired by human-like open-world learning, the authors propose a two-stage discovery process: 1) Compositional Perception, which maps unstructured tokens to a reusable vocabulary of primitives through primitive competition and evidence consolidation; and 2) Generalizing Induction, which leverages the induced geometric structure and applies geometric calibration based on the principles of proximity and continuity. CoGe-GCD is designed as a plug-and-play inductive-bias module between backbone and projection head, allowing integration into various existing GCD frameworks. Across standard fine- and coarse-grained benchmarks, the module delivers consistent improvements in overall and novel-category accuracy and improves unknown-class number estimation with marginal overhead.

All reviewers reach to a positive consensus on the motivation and the two-stage human-like approach: “the idea of introducing CoGe-GCD as a plug and play module to be a novel contribution” (Reviewer oxwB); “The intuition of modeling human-like compositional reasoning process makes sense” (Reviewer A1sc); “The paper introduces human-like reasoning (reusing known primitives and inducing new combinations) into the GCD task, providing an insightful perspective for open-world category discovery” (Reviewer g2tn); “This decomposition into Compositional Perception and Generalizing Induction gives a clear missing capability story and makes the paper easy to evaluate on its own terms” (Reviewer VZ29). The plug-and-play-styled module is also praised by Reviewers g2tn and VZ29 for its versatility and practicality.

Most concerns were around ablation studies, datasets, comparison methods and results, presentation quality and readability, which have been addressed in the rebuttal stage. Reviewers oxwB, g2tn and VZ29 have acknowledged that their concerns have been addressed and are inclined to accept the paper. While Reviewer A1sc still has unsolved concerns regarding Significance of GCD, Experimental protocol of GCD, and isualizations and Supplemental Code.

I agree with most of reviewers on the proposed two-stage approach to mimic human-like compositional reasoning process. I also appreciate that the manifold assumption and the graph Laplacian regularization widely used in the era of traditional machine learning was introduced. But it is unclear that such a manifold assumption holds. The authors are expected to demonstrate the manifold assumption holds for tokens. The authors also should try to address the unsolved concerns to Reviewer A1sc.